# Proposal for realizing anomalous Floquet insulators via Chern band annihilation

Carolyn Zhang[1], Tobias Holder[2], Netanel H. Lindner[3], Mark S. Rudner[4] and Erez Berg[2]

**1** Department of Physics, Kadanoff Center for Theoretical Physics,
University of Chicago, Chicago, Illinois 60637, USA
**2** Department of Condensed Matter Physics,
Weizmann Institute of Science, Rehovot 7610001, Israel
**3** Physics Department, Technion, Haifa 320003, Israel
**4** Center for Quantum Devices and Niels Bohr International Academy,
University of Copenhagen, 2100 Copenhagen, Denmark

## Abstract

Two-dimensional periodically driven systems can host an unconventional topological phase unattainable for equilibrium systems, termed the Anomalous Floquet-Anderson insulator (AFAI). The AFAI features a quasi-energy spectrum with chiral edge modes and a fully localized bulk, leading to non-adiabatic but quantized charge pumping. Here, we show how such a Floquet phase can be realized in a driven, disordered Quantum Anomalous Hall insulator, which is assumed to have two critical energies where the localization length diverges, carrying states with opposite Chern numbers. Driving the system at a frequency close to resonance between these two energies localizes the critical states and annihilates the Chern bands, giving rise to an AFAI phase. We exemplify this principle by studying a model for a driven, magnetically doped topological insulator film, where the annihilation of the Chern bands and the formation of the AFAI phase is demonstrated using the rotating wave approximation. This is complemented by a scaling analysis of the localization length for two copies of a quantum Hall network model with a tunable coupling between them. We find that by tuning the frequency of the driving close to resonance, the driving strength required to stabilize the AFAI phase can be made arbitrarily small.

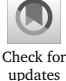
---

# 1 Introduction

Periodic driving of quantum systems has opened exciting new avenues for realizing topological phases [1–7]. Notably, Floquet driving has been utilized to obtain dynamical analogues of stationary topological phases [8–27]. In such driven systems, the time evolution over a driving period $T$, implemented by the Floquet unitary operator $U_F(T)$, can be accurately described by the evolution of a stationary, spatially local effective Hamiltonian $H_{\text{eff}}$ such that $U_F = e^{-iH_{\text{eff}}T}$. These Floquet phases have been observed in a variety of experiments [1, 28–36].

However, Floquet driving can also produce genuinely new phases that do not occur in stationary settings [9, 37–58]. One example of such a phase was presented in Ref. [40], where a clean, non-interacting two-dimensional (2D) model was shown to host chiral edge states, despite the fact that all the bulk bands carry zero Chern numbers. In a stationary setup, this would be impossible because the topology of the bulk bands, given by their Chern numbers, completely determines the edge properties.

The role of disorder in such 2D "anomalous" topological Floquet phases, with vanishing Chern numbers, was first studied in Ref. [50]. There, it was shown that spatial disorder localizes all bulk Floquet states, while the chiral edge states remain robust. The driven phase that emerges in such a system, coined the Anomalous Floquet Anderson Insulator (AFAI), displays chiral edge states at *all* quasi-energies [50, 58, 59]; the net number of chiral edge states is given by the value of a single winding number, $\mathcal{W}$. In contrast, in stationary systems the existence of a chiral edge state necessitates delocalization of bulk states at certain energies [60]. For example, in quantum anomalous Hall (QAH) systems, there must be a single energy near the middle of each Chern band where the localization length diverges [61, 62]. The AFAI therefore exhibits properties that cannot be realized without periodic driving.

In this work we propose a method to realize an AFAI phase in a solid state system. The idea is to start from a disordered QAH material, and apply a periodic driving field that resonantly couples the delocalized states in two Chern bands with opposite Chern numbers. We argue that such driving localizes the states, and generally leads to the formation of an AFAI phase, independent of many of the microscopic details of the system and the properties of the driving

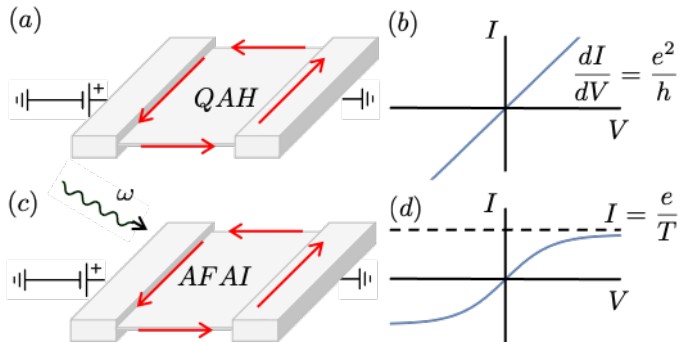

Figure 1: (a) The stationary QAH phase demonstrates a chiral edge state. Therefore, the slope of the $I - V$ curve, i. e. the conductivity is quantized to $\sim e^2/h$ for small voltage, with only minor corrections at higher voltage. (b) In the AFAI the current behaves in the same way at small bias, but it saturates to $e/T$ at higher voltage, leading to a quantized current [50, 58].

field. The resulting AFAI phase displays quantized transport properties that are different from those of the initial QAH phase (see Fig. 1).

This paper is organized as follows. In Sec. 2, we summarize the physical picture that underlies this work. We also summarize our main results obtained from two approaches: a concrete Hamiltonian model and a disordered network model. In Sec. 3, we elaborate on the Hamiltonian model, which describes a magnetically doped topological insulator film realizing a QAH phase. We obtain the energies of the delocalized states in the stationary system using the self-consistent Born approximation and make physical arguments for the qualitative features of the phase diagram in the presence of driving field. Next, in Sec. 4 we introduce a bilayer network model representing two Chern bands coupled by a nearly-resonant drive. Using this network model, we obtain a similar phase diagram to that of the Hamiltonian model, and also determine the critical exponents associated with the phase transitions. Additional technical details are presented in the appendices.

## 2 Physical Picture

A QAH insulator is marked by a quantized Hall conductivity in the absence of a magnetic field, typically due to magnetic polarization and spin-orbit coupling [63, 64]. Starting from a model of a simple QAH system with two Chern bands, adding disorder generically localizes all bulk states except for those at critical energies $\epsilon_1$ and $\epsilon_{-1}$ in the $C = 1$ and $C = -1$ Chern bands, respectively [62] (see Fig. 2a). Tuning the Fermi energy through $\epsilon_{\pm 1}$ results in $\pm \frac{e^2}{h}$ quantized jumps in the Hall conductivity. The QAH effect occurs when the Fermi energy lies between $\epsilon_1$ and $\epsilon_{-1}$.

Suppose we apply a harmonic driving field at $\omega = \epsilon_{-1} - \epsilon_1$. This field resonantly couples states that carry opposite Chern numbers. The Floquet spectrum can then be obtained from the extended Hamiltonian $H_F$, given by the infinite block-tridiagonal matrix:

$$
H_F = \begin{pmatrix} \ddots & \vdots & \vdots & \iddots \\ \cdots & H + \omega & H^{(-1)} & \cdots \\ \cdots & H^{(1)} & H & \cdots \\ \iddots & \vdots & \vdots & \ddots \end{pmatrix},
\tag{1}
$$

where $H$ is the QAH Hamiltonian in the absence of the driving. Each block on the diagonal

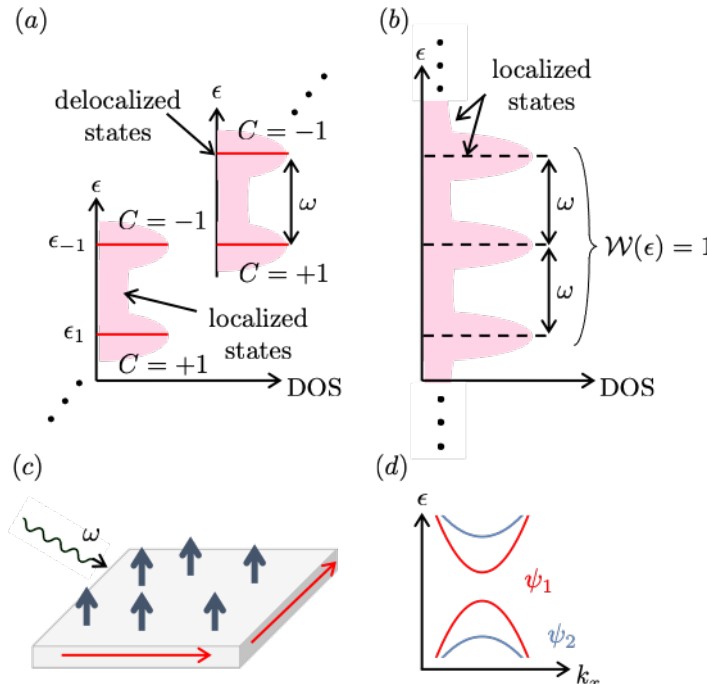

Figure 2: (a) Schematic density of states as a function of energy for a disordered QAH insulator. Anticipating a periodic driving with frequency $\omega$, we show two consecutive Floquet zones. The spectrum consists mostly of localized states, but has delocalized states at energies $\epsilon_{\pm 1}$ within the Chern bands with $C = \pm 1$. (b) Density of states of the Floquet spectrum obtained when the system is driven at the resonant frequency $\omega = \epsilon_{-1} - \epsilon_{+1}$. The driving couples the delocalized states from the two Chern bands and causes them to localize, so that all bulk states become localized. The winding number $\mathcal{W}(\epsilon)$ at every quasi-energy $\epsilon$ is equal to 1. (c) Schematic of the thin film ferromagnetic TI, forming a QAH state (Section 3). The blue arrows represent the spin polarization and the red arrow around the edge of the film represents the chiral edge state. To this stationary system we add a driving field at frequency $\omega$. (d) The spectrum of surface states is hybridized between the top and bottom of the film. In the absence of disorder, the surface states are described by two massive Dirac fields $\psi_1$ and $\psi_2$.

of the extended Hamiltonian $H_F$ acts on a different Fourier harmonic component $|\phi_l\rangle$ of the Floquet state $|\psi(t)\rangle$:

$$|\psi(t)\rangle = e^{-i\epsilon t} \sum_l e^{-il\omega t}|\phi_l\rangle. \tag{2}$$

The matrices $H^{(1)}$ and $H^{(-1)} = H^{(1)\dagger}$, proportional to the driving amplitude $A$, describe transitions accompanied by the absorption and emission of a single photon from the driving field. $H_F$ produces physically equivalent eigenstates at quasienergies $\epsilon + n\omega$ where $\epsilon \in (-\frac{\omega}{2}, \frac{\omega}{2}]$ and $n \in \mathbb{Z}$ indicates the *Floquet zone*.

We can approximate the Floquet eigenstates by truncating $H_F$ to a finite number of harmonics. For small $\frac{A}{\omega}$, it is sufficient to only include $l = 0, 1$, since this captures all the states that are resonantly coupled to first order in driving field.

If the driving frequency exactly satisfies $\omega = \epsilon_{-1} - \epsilon_1$, then the delocalized states at energies $\epsilon_{\pm 1}$ are resonantly coupled. Since these states carry opposite Chern numbers, the drive-induced resonant coupling causes them to "annihilate", and become localized. For perfectly resonant coupling, one may expect that an arbitrarily small driving amplitude is sufficient to localize

all the bulk Floquet eigenstates (Fig. 2b). If the frequency is detuned from the resonance, a non-zero minimum driving amplitude is required to achieve complete localization.

We argue that, if the bulk states are all localized, the resulting phase is an AFAI. To see this, consider a system with open boundary conditions. The chiral edge states of the QAH system at energies $\epsilon_1 < \epsilon < \epsilon_{-1}$ cannot become localized as long as the driving amplitude is sufficiently small compared to $\omega$. Since all the bulk states are now localized, the edge states cannot terminate at any quasienergy, and must persist over the entire Floquet zone. In other words, the winding number $\mathcal{W}(\epsilon) = 1$ for all $\epsilon$. This is the defining characteristic of the AFAI phase [50].

Note that while we study Chern band annihilation by tuning the drive frequency and amplitude with time-independent disorder, a similar phenomenon was studied in Ref. [65] by tuning periodically-modulated disorder. It was shown there that for a stationary model with two Chern bands with energy separation $\epsilon_{-1} - \epsilon_1$, introducing random on-site potential disorder with frequency $\omega$ causes the Chern bands to annihilate, driving the system into either an Anderson insulator phase or an AFAI phase. The deciding factor is the ratio between the gap around $\epsilon = 0$, which is $\epsilon_{-1} - \epsilon_1$, and the gap around $\epsilon = \frac{\omega}{2}$, which is $\omega - (\epsilon_{-1} - \epsilon_1)$: if the gap around $\epsilon = 0$ is smaller, then disorder leads to an Anderson insulating phase, while if the gap around $\epsilon = \frac{\omega}{2}$ is smaller, then disorder leads to an AFAI. The critical disorder amplitude for the AFAI transition depends on the size of the gap around $\epsilon = \frac{\omega}{2}$ and becomes infinitesimal as this gap closes. We focus in this work on the limit $\omega = \epsilon_{-1} - \epsilon_1$, where the gap around $\epsilon = \frac{\omega}{2}$ closes, and our results are complementary to those of Ref. [65]. We find that the critical amplitude for the spatially uniform Floquet drive depends on the size of this gap, which we call the detuning from resonance, and becomes infinitesimal as this gap closes, which is the condition for driving on resonance.

To confirm the idea of Chern band annihilation outlined above, we use two approaches that give complementary results.

## 2.1 Hamiltonian Model

First, we study a minimal Hamiltonian model of the QAH insulator for which we can reliably compute the delocalization energies $\epsilon_{\pm 1}$. The model describes a QAH system formed in a magnetically doped thin topological insulator film [63], subjected to a time-periodic perpendicular electric field.

In order to determine the phase diagram of the driven system, we first find the delocalization energies $\epsilon_{\pm 1}$ in the limit of zero driving. According to the renormalization group (RG) treatment of the QH plateau transition [61,66–68], the delocalization energies can be found in the weak disorder limit by computing the conductivity tensor perturbatively in the disorder strength, within the self-consistent Born approximation (SCBA). The values of longitudinal and Hall conductivities, $\sigma_{xx}$ and $\sigma_{xy}$, computed perturbatively, then serve as the initial conditions for the RG flow. The flow diagram [66,67], shown schematically in Fig. 3, contains stable fixed points that correspond to QH insulators (where $\sigma_{xx} = 0$ and $\sigma_{xy}$ is quantized in units of $e^2/h$), and unstable fixed points that describe the plateau transitions. While Pruisken's RG analysis was originally developed to describe the integer QH plateau transition, the transitions between different QAH phases belong to the same universality class [62]. Within this treatment, the plateau transitions occur when $\sigma_{xy} = (n + \frac{1}{2})\frac{e^2}{h}$ with $n \in \mathbb{Z}$. Within our model, we locate the energies where delocalized states carrying non-zero Chern numbers occur by computing $\sigma_{xy}$ as a function of the Fermi energy within the SCBA, and finding the energies where $\sigma_{xy} = e^2/(2h)$. We henceforth measure the conductivity in units of $e^2/h$.

The SCBA computation of the semiclassical conductivity requires some care [69–71]. For our simple model, we were able to compute the Hall conductivity and obtain $\epsilon_{\pm 1}$ as a function

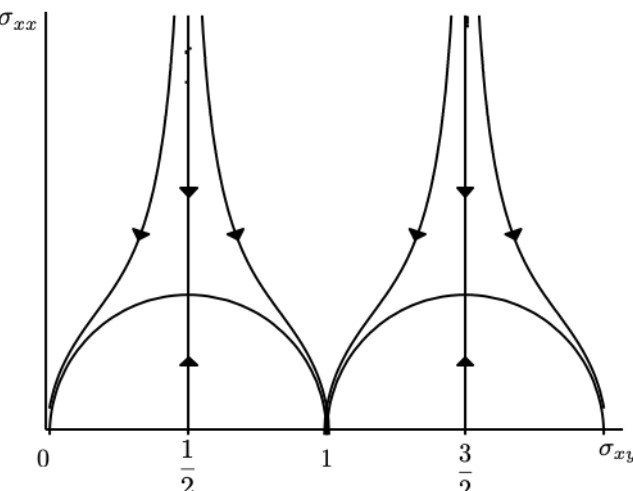

Figure 3: Schematic RG flow diagram for the integer QH effect in terms of the longitudinal and Hall conductivities. The effect of disorder increases with decreasing $\sigma_{xx}$, ultimately driving the system into a state with quantized Hall conductivity. The critical line at $\sigma_{xy} = \frac{1}{2}$ separates states that flow to $\sigma_{xy} = 0$ and $\sigma_{xy} = 1$. Reproduced from Refs. [66, 67].

of Hamiltonian parameters in the intrinsic metallic regime [71], where the disorder potential is assumed to be weak and has a Gaussian distribution.

We then turn to study the phase diagram of the driven system. We set $\omega = \epsilon_{-1} - \epsilon_1$ and infer the phase diagram starting from the limit of zero driving amplitude, where we find regions of different $\mathcal{W}$ separated by critical lines. Our analysis strongly suggests that the driving stabilizes the AFAI phase, even if the drive frequency is not exactly resonant with $\epsilon_{-1} - \epsilon_1$.

## 2.2 Network Model

In order to explore the universal aspects of Chern band annihilation due to the driving-induced coupling of states with opposite Chern numbers, we construct a disordered Chalker-Coddington type network model [72]. The Chalker-Coddington network model describes transport in a lattice with fixed (non-random) scattering matrices at nodes and random phase matrices on links. In the past, different kinds of network models have been used successfully to study QH localization-delocalization transitions and their variants [72–83]. Network models are particularly useful because they can be used to efficiently compute the localization length on a quasi-1D geometry, i.e., a long cylinder [79, 84–86].

The network model that we construct consists of two 2D "layers" with opposite chirality, to represent the QAH critical states with Chern numbers $\pm 1$ that occur at the same quasienergy. We emphasize that the two layers do not correspond to two different spatial locations, but rather to the states near the two delocalization energies $\epsilon_{\pm 1}$ (Fig. 2), brought close to resonance by the driving. The scattering between the two layers then corresponds to the nearly resonant driving-induced coupling, which will be parametrized by a strength $J_p$.

Numerically, we find a phase diagram which gives the winding number $\mathcal{W}$ as a function of the energy in each of the two layers (identified as a parameter that tunes each layer through its critical point), and as a function of $J_p$. We explain how to define the winding number in the network model in Sec. 4.3. From finite size scaling, we find that at resonance the localization length scales as $\xi \sim J_p^{-\nu_p}$, with $\nu_p \simeq 4.2$. Comparing to the known value for the correlation length exponent of the QH transition within a single layer, $\nu \simeq 2.6$ [72, 87–89], this implies that the inter-layer coupling operator from the Floquet driving is relevant, albeit less relevant than

the operator that tunes the QH transition. When the frequency is at resonance, an arbitrarily weak drive brings the system into a localized AFAI phase. Away from resonance, we find that the driving strength needs to exceed a critical value that depends on the detuning in order to localize all the states. These findings are in qualitative agreement with those of the Hamiltonian model described above.

# 3 Driven QAH System

In this section, we study a Hamiltonian model of the experimental setup depicted in Fig. 2. Our model describes a QAH system constructed from a thin film of a ferromagnetically doped TI, as introduced in Ref. [63] and experimentally realized in Refs. [90–93]. The QAH insulator can also be realized in other systems, such as in twisted bilayer graphene [94].

The model includes two Dirac modes that reside on the opposite surfaces of the film. The ferromagnetic moment, pointing in the direction perpendicular to the film (which we denote by $\hat{z}$), results in a mass term for the two Dirac modes, of magnitude $\Delta$. There is also tunneling between the two surfaces of strength $m(k) = m_0 + Bk^2 = m^*(k)$, where $k = |\mathbf{k}|$ is the magnitude of the momentum parallel to the film. The stationary (undriven) Hamiltonian takes the simple form:

$$
\tilde{H}_{\text{QAH}} = \begin{pmatrix} v_F\left(k_y\sigma_x - k_x\sigma_y\right) + \Delta\sigma_z & m(k) \\ m(k) & v_F\left(k_x\sigma_y - k_y\sigma_x\right) + \Delta\sigma_z \end{pmatrix},
\tag{3}
$$

where $\sigma_{x,y,z}$ act in spin space. Here $\tilde{H}_{\text{QAH}}$ is written in the ordered basis $\{|t\uparrow\rangle, |t\downarrow\rangle, |b\uparrow\rangle, |b\downarrow\rangle\}$, where $t$ and $b$ label the top and bottom surfaces respectively. We can bring $\tilde{H}_{\text{QAH}}$ into a block diagonal form by performing a unitary transformation into the bonding/anti-bonding basis $\{|+\uparrow\rangle, |-\downarrow\rangle, |+\downarrow\rangle, |-\uparrow\rangle\}$, where $+$ and $-$ correspond to the bonding and anti-bonding combinations of states on the two surfaces, respectively [1]:

$$
H_{\text{QAH}} = \begin{pmatrix} h(k) + \Delta\sigma_z & 0 \\ 0 & h(k) - \Delta\sigma_z \end{pmatrix},
\tag{4}
$$

where $h(k) = m(k)\sigma_z + v_F(k_y\sigma_x - k_x\sigma_y)$. In this basis, the system consists of two decoupled Dirac fields $\psi_1$ and $\psi_2$ with mass terms $m_1(k) = m(k) - \Delta$ and $m_2(k) = m(k) + \Delta$, respectively. We further denote $m_1(0) = m_1$ and $m_2(0) = m_2$.

We now compute the Hall conductivity for this system in the presence of Gaussian-distributed, $\delta$-correlated potential disorder. Our goal is to find the critical energy where $\sigma_{xy} = \frac{1}{2}$, as a function of the system's parameters. We then consider the effects of driving at the resonance frequency $\omega = \epsilon_{-1} - \epsilon_{+1}$ within the rotating wave approximation, using the effective time independent Hamiltonian obtained from truncating Eq. (1).

## 3.1 Calculation of Hall conductivity

We now determine the critical lines of the stationary Hamiltonian given by Eq. (4). Because Dirac modes $\psi_1$ and $\psi_2$ are approximately decoupled, the total Hall conductivity is simply the sum of the Hall conductivities due to $\psi_1$ and $\psi_2$:

$$
\sigma_{xy} = \sigma_{xy}^{(1)} + \sigma_{xy}^{(2)}.
\tag{5}
$$

---

[1]The transformation reads explicitly $(|t\uparrow\rangle \pm |b\uparrow\rangle)\sqrt{2}, (|t\downarrow\rangle \pm |b\downarrow\rangle)\sqrt{2}$

Without loss of generality, we choose $m_0 > 0, \Delta > 0$ and $\Delta$ close to $m_0$ so that $|m_1(k)| \ll |m_2(k)|$ for the small values of $k$ consistent with the low energy limit ($k \ll m_2/v_F$).

The energy dispersion of the conduction band of mode $\psi_1$ is given by $\epsilon_k = \sqrt{(v_F k)^2 + (m_0 - \Delta + Bk^2)^2}$. We consider an electron doped system with Fermi energy $\epsilon_F = \epsilon_{k_F}$ and Fermi momentum $k_F$.

In the following, we calculate the critical lines in the plane spanned by $\Delta$ and energy $\epsilon$ for a given value of $m_0$. We could proceed in the same way for the critical line in the $m_0 - \epsilon$ plane for a given $\Delta$. When the energy $\epsilon$ is between $m_1$ and $m_2$, it lies within the gap of the field $\psi_2$. For these values of $\epsilon$, $\sigma_{xy}^{(2)}$ is quantized and given solely by the intrinsic (Berry curvature) contribution $\sigma_{xy,0}^{(2)}$, with

$$\sigma_{xy,0}^{(2)} = \frac{\text{sgn}(m_0 + \Delta)}{2}. \tag{6}$$

Taking $m_0 > 0, \Delta > 0$, this gives a constant value of $\sigma_{xy,0}^{(2)} = \frac{1}{2}$. In order for the total semiclassical Hall conductivity to be a half integer, which is the condition for the entire system to be critical (as discussed in Sec. 2.1), it is therefore required that the contribution of the field $\psi_1$ be $\sigma_{xy}^{(1)} = 0$. Note that the Fermi energy is inside the band of $\psi_1$, so that the semiclassical value of $\sigma_{xy}^{(1)}$ includes non-quantized contributions. The transverse dc-conductivity for a clean system is given by the Kubo formula,

$$\tilde{\sigma}_{xy}^{(1)} = \lim_{\omega \to 0} \frac{2\pi}{\omega} \text{Tr} \int \frac{d\epsilon}{2\pi} \frac{d^2k}{(2\pi)^2} j_x G_0(\epsilon + \omega, \mathbf{k}) j_y G_0(\epsilon, \mathbf{k}), \tag{7}$$

where $G_0(\epsilon, \mathbf{k})$ is the causal (time-ordered) Green's function of the effective two-band Hamiltonian $H_1 = v_F(k_x \sigma_y + k_y \sigma_x) + m_1(k)\sigma_z$ describing the field $\psi_1$. The current operator is $j_i = \partial H_1/\partial k_i$. Taking the limit $\omega \to 0$ leads to the Kubo-Streda formula of conductivity [95], which contains both a contribution from all filled states below the Fermi energy and a piece from the Fermi energy itself. To find the disorder average of $\sigma_{xy}^{(1)}$ within the ladder approximation [70], we (1) replace $G_0(\epsilon, \mathbf{k})$ by the disorder-averaged Green's function $G(\epsilon, \mathbf{k})$ calculated within the SCBA, and (2) replace $j_x$ by the renormalized vertex $\Upsilon_x$:

$$\sigma_{xy}^{(1)} = \lim_{\omega \to 0} \frac{2\pi}{\omega} \text{Tr} \int \frac{d\epsilon d^2k}{(2\pi)^3} \Upsilon_x(\epsilon, \mathbf{k}) G(\epsilon + \omega, \mathbf{k}) j_y G(\epsilon, \mathbf{k}), \tag{8}$$

where $G(\epsilon, \mathbf{k}) = (G_0^{-1}(\epsilon, \mathbf{k}) - \Sigma)^{-1}$ is the SCBA Green's function including the self-energy $\Sigma$ due to impurity scattering.

Details of the evaluation of Eq. (8) can be found in the Appendix A. We find that, at low energies, $\sigma_{xy}^{(1)}$ vanishes when the Fermi energy satisfies

$$\Delta = m_0 + \frac{21 B \epsilon_F^2}{16 v_F^2}. \tag{9}$$

This result is easily generalized for $|m_2| \ll |m_1|$ by interchanging $\psi_1$ and $\psi_2$. In this case it follows analogously that $\sigma_{xy}^{(1)} = \sigma_{xy,0}^{(1)} = -\frac{1}{2}$ and the critical line is found for $\sigma_{xy}^{(2)} = 0$, which in turn evaluates to the condition $\Delta = -m_0 - 21B\epsilon_F^2/16v_F^2$. A representative phase diagram in the $\Delta - \epsilon$ plane is shown in Fig. 4.

Depending on the relative signs of $m_0$ and $B$, the critical lines curve either away from the origin, $\Delta = 0$, or toward $\Delta = 0$ as $\epsilon$ moves away from 0. For concreteness, we assume that $m_0 B > 0$, such that the critical lines curve away from the origin. In a given system $m_0 B$ can be of either sign, depending on the microscopics of the material. Note that for $B = 0$, the critical curves are vertical lines positioned at $\Delta = \pm m_0$. In other words, when the mass term is

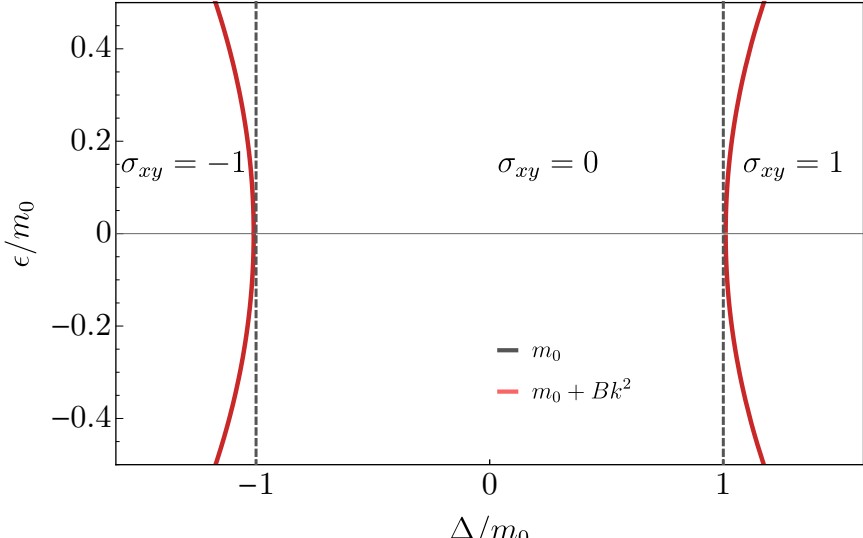

Figure 4: Phase diagram for the stationary system described by Eq. (3) in the presence of weak disorder. The dashed, gray lines denote the energy where the semiclassical value of $\sigma_{xy}$ equals $\pm\frac{1}{2}$ for $m(k) = m_0$ and the bold red lines denote the critical energies for a disordered system, [Eq. (9)]. The Hall conductivity indicated in each region refers to the Hall conductivity at the RG fixed point. $B = 0.5v_F^2/m_0$ was used in the figure.

not $k$-dependent, states at $\Delta = \pm m_0$, where the magnetization and tunnel coupling between the two surfaces are of equal strength, are delocalized at all energies, and all other states are localized.

For $B \neq 0$ and $\Delta > m_0$, there are always two distinct solutions for $\epsilon_F$ in Eq. (9). These two solutions correspond to $\epsilon_{+1}$ and $\epsilon_{-1}$, the two delocalization energies. This gives the frequency $\omega = \epsilon_{-1} - \epsilon_{+1}$ at which we drive the QAH system to realize the AFAI phase via Chern band annihilation.

## 3.2 Driving the QAH Insulator

We now consider driving the system with a time-dependent electric field perpendicular to the film, corresponding to the following perturbation to the Hamiltonian:

$$H_{A_0} + H_{A_1} = \begin{pmatrix} 0 & A_0\sigma_x + 2A_1\cos(\omega t)\sigma_x \\ A_0\sigma_x + 2A_1\cos(\omega t)\sigma_x & 0 \end{pmatrix}. \tag{10}$$

Here, $A_0$ specifies a fixed potential difference between the two surfaces of the film, while $A_1$ is a periodically modulated potential difference with frequency $\omega$. We note that Eq. (10) is written in the same bonding/anti-bonding basis as Eq. (4), which leads to the off-diagonal structure of the inter-layer potential. To leading order in $A_1/\omega$, we can truncate the extended Hamiltonian $H_F$ in Eq. (1) to include just two harmonics, as discussed in Sec. 2. In this way, we obtain

$$H_{\text{eff}} = \begin{pmatrix} H + \omega & H_{A_1} \\ H_{A_1} & H \end{pmatrix}, \tag{11}$$

where $H = H_{\text{QAH}} + H_{A_0}$. The corresponding action for such a system can be written as

$$S_{\text{driven}} = S_0 + S_\omega + S_{0,\omega}, \tag{12}$$

where $S_0$ describes the fields $\psi_1$, $\psi_2$ in the zeroth Floquet zone, $S_\omega$ describes the fields $\psi_{1,\omega}$, $\psi_{2,\omega}$ in the first Floquet zone, and $S_{0,\omega}$ describes the coupling between them. The action $S_0$ is given by

$$
S_0 = \int \frac{d^2k}{(2\pi)^2}\Big[\overline{\psi}_1(\epsilon - v_F(k_x\sigma_y + k_y\sigma_x) - m_1(k)\sigma_z)\psi_1
$$
$$
+ \overline{\psi}_2(\epsilon - v_F(-k_x\sigma_y + k_y\sigma_x) - m_2(k)\sigma_z)\psi_2
$$
$$
- \overline{\psi}_1 A_0\sigma_x\psi_2 - \overline{\psi}_2 A_0\sigma_x\psi_1\Big],
\tag{13}
$$

while $S_\omega$ follows from $S_0$ by the replacement $\epsilon \to \epsilon - \omega$ and $\psi_1, \psi_2 \to \psi_{1,\omega}, \psi_{2,\omega}$. The coupling term is given by

$$
S_{0,\omega} = -A_1 \int \frac{d^2k}{(2\pi)^2}\Big[\overline{\psi}_1\sigma_x\psi_{2,\omega} + \overline{\psi}_2\sigma_x\psi_{1,\omega} + \overline{\psi}_{1,\omega}\sigma_x\psi_2 + \overline{\psi}_{2,\omega}\sigma_x\psi_1\Big].
\tag{14}
$$

Note that in both Eqs. (13) and (14), $H_{A_0} + H_{A_1}$ couples $\psi_1$ to $\psi_2$ and $\psi_{2,\omega}$ (but not to $\psi_{1,\omega}$), and $\psi_{1,\omega}$ to $\psi_2$ and $\psi_{2,\omega}$. This is due to $\psi_1$ being even under mirror symmetry through the middle of the film, while $\psi_2$ is odd; hence a perpendicular electric field can only couple fields 1 and 2.

Because $\psi_2$ and $\psi_{2,\omega}$ have large masses compared to $\psi_1$ and $\psi_{1,\omega}$, we can integrate them out to obtain an action for the light fields only. This yields

$$
S'_{\text{driven}} = S'_0 + S'_\omega + S'_{0,\omega},
\tag{15}
$$

where

$$
S'_0 = \int \frac{d^2k}{(2\pi)^2}\Big[\overline{\psi}_1(\epsilon - v_F(k_x\sigma_y + k_y\sigma_x) - m_1\sigma_z)\psi_1
$$
$$
- \overline{\psi}_1\sigma_x\big(A_0^2 G_2(\epsilon, \boldsymbol{k}) + A_1^2 G_{2,\omega}(\epsilon - \omega, \boldsymbol{k})\big)\sigma_x\psi_1\Big],
\tag{16}
$$

and $S'_\omega$ again follows by the replacement $\epsilon \to \epsilon - \omega$ and $\psi_1 \to \psi_{1,\omega}$. Finally,

$$
S'_{0,\omega} = -2A_0 A_1 \int \frac{d^2k}{(2\pi)^2}\Big[\overline{\psi}_1\sigma_x\big(G_2(\epsilon, \boldsymbol{k}) + G_{2,\omega}(\epsilon - \omega, \boldsymbol{k})\big)\sigma_x\psi_{1,\omega}
$$
$$
+ \overline{\psi}_{1,\omega}\sigma_x\big(G_2(\epsilon - \omega, \boldsymbol{k}) + G_{2\omega}(\epsilon, \boldsymbol{k})\sigma_x\psi_1\Big].
\tag{17}
$$

By construction, this description amounts to a low energy theory, with a UV cutoff $K$ for the $k$-integration of size $K \sim m_2/v_F$. We assume that $BK \ll v_F$, so at large energies near the limits of integration, the dispersion remains approximately linear. Crucially, in Eq. (17), a coupling between $\psi_1$ and $\psi_{1,\omega}$ has been generated. To evaluate this coupling, we employ the retarded Green's function

$$
G_2(\epsilon, \boldsymbol{k}) = \frac{\epsilon + i0^+ + v_F(-k_x\sigma_y + k_y\sigma_x) + m_2(k)\sigma_z}{(\epsilon + i0^+)^2 - v_F^2|k|^2 - m_2(k)^2},
\tag{18}
$$

and $G_{2,\omega}(\epsilon, \boldsymbol{k}) = G_2(\epsilon - \omega, \boldsymbol{k})$. To lowest order in $\frac{v_F k_F}{m_2}$ and $\frac{\epsilon}{m_2}$, the masses of $\psi_1$ and $\psi_{1,\omega}$ in Eq. (16) are renormalized according to $m_1 \to m_1 + \frac{A_0^2 + A_1^2}{m_2}$ and the coupling between $\psi_1$ and $\psi_{1,\omega}$ in Eq. (17) is given by $\frac{2A_0 A_1}{m_2}\sigma_z$.

In summary, the $A_0$ and $A_1$ terms affect $\psi_1$ and $\psi_{1,\omega}$ in two ways: (1) they shift $m_0$ by a constant, thereby shifting the critical lines and (2) they add a $\sigma_z$ coupling between $\psi_1$ and

$\psi_{1,\omega}$. To maximize the coupling while minimizing the shift in $m_0$, we choose $A_0 = A_1 = A$. In this case, the effective Hamiltonian for $\psi_1$ and $\psi_{1,\omega}$ in the driven system is given by

$$H'_{\text{eff}} = \begin{pmatrix} M(k)\sigma_z + h_{\boldsymbol{k}} + \omega & \frac{2A^2}{m_2}\sigma_z \\ \frac{2A^2}{m_2}\sigma_z & M(k)\sigma_z + h_{\boldsymbol{k}} \end{pmatrix}, \tag{19}$$

where $M(k) = m_1(k) - 2A^2/m_2$ and $h_{\boldsymbol{k}} = v_F(k_x\sigma_y + k_y\sigma_x)$.

We first discuss the phase diagram of Eq. (19) without the off-diagonal blocks. In this case, the positions of the new critical lines in the $(\Delta, \epsilon)$ plane in the presence of weak potential disorder follow from Eq. (9) with the replacement $m_0 \to m_0 + \frac{2A^2}{m_2}$. The critical line that corresponds to the lower right block of Eq. (19) is given by

$$\Delta = m_0 + \frac{2A^2}{m_0 + \Delta} + \frac{21B\epsilon^2}{16v_F^2}, \tag{20}$$

where we substituted $m_2 = m_0 + \Delta$. The critical line corresponding to the upper left block of Eq. (19) is given by Eq. (20) where $\epsilon$ is replaced by $\epsilon - \omega$.

Solving Eq. (20) for $\epsilon$ gives

$$\epsilon_c(\Delta) = \sqrt{\frac{v_F^2[-2A^2 - (m_0 - \Delta)(m_0 + \Delta)]}{(21B/16)(m_0 + \Delta)}}, \tag{21}$$

where $\epsilon_c(\Delta)$ is the critical energy in the lower block. Eq. (21) determines the resonant driving frequency $\omega = 2\epsilon_c(\Delta)$. The delocalization lines for $\psi_1$ and $\psi_{1,\omega}$ without the off-diagonal coupling terms are shown by the red line in Fig. 5. The figure also shows the value of the winding number $\mathcal{W}$ in the different regions separated by the critical lines.

Away from the resonance point at $\epsilon = \frac{\omega}{2}$, the off-diagonal term in Eq. (19) should not change the delocalization lines significantly: away from this energy the delocalized states are not strongly hybridized. Near resonance, however, this can no longer be assumed. The hybridization invalidates Eq. (20) near these energies because Eq. (20) only holds for the simple two-band Hamiltonian $M(k) + h_{\boldsymbol{k}}$. In order to determine the delocalization lines in the presence of the coupling, it becomes necessary to compute the SCBA Hall conductivity for the four-band model in Eq. (19) with added Gaussian $\delta$-correlated potential disorder in $\psi_1$ and $\psi_{1,\omega}$.

We do not perform this calculation here. Instead we infer qualitatively how the hybridization may change the delocalization lines, using the values of $\mathcal{W}$ in the different regions separated by the red lines in Fig. 5. One possibility is that the driving localizes the states at the crossing point of the two critical lines, leaving all the states for that value of $\Delta$ completely localized. The new critical lines are shown schematically by the dashed blue lines in Fig. 5. In that case, the system is in the AFAI phase, since $\mathcal{W} = 1$ for all quasi-energies. Another possibility is that the crossing of the critical lines may shift as a result of the off-diagonal coupling. (Note that if the delocalization lines were reconnected above and below $\epsilon = \frac{\omega}{2}$, then the $\mathcal{W} = 0$ region would be connected to the $\mathcal{W} = 2$ region, which cannot happen because regions with different $\mathcal{W}$ must be separated by critical lines.) To determine which of these possibilities is realized, we must perform a more detailed calculation. This is done in the next Section, where we show that an AFAI phase is indeed realized generically for nearly-resonant driving.

## 4 Localization of critical states by resonant driving: Network model

We now consider the fate of the states in the vicinity of the crossing point of the critical energies in Fig. 5. These critical states carry opposite Chern numbers, as can be seen from the jumps in $\mathcal{W}$
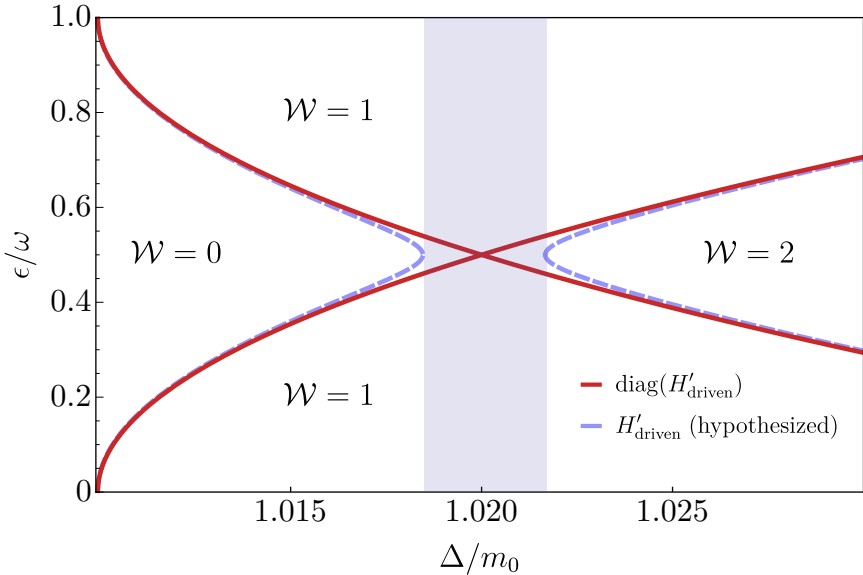

Figure 5: Phase diagram for the driven system as a function of $\Delta$ and the quasienergy $\epsilon$ at a fixed driving frequency $\omega$, in the presence of random potential disorder. The red lines show the delocalized energies obtained by setting the off-diagonal blocks in Eq. (19) to zero. The blue dashed lines schematically indicate the expected locations of the delocalized quasienergies for $H'_{\text{eff}}$ [Eq. (19)] including the off-diagonal blocks. In this case, the blue shaded region is in the AFAI phase, as can be seen from the fact that in this region, $\mathcal{W} = 1$ for all quasi-energies. For a material with a fixed value of $\Delta$ and $m_0$, the resonant driving frequency can be obtained via Eq. (21). The following parameters were used in the figure: $A_0 = A_1 = 0.1m_0$, $\omega = 0.24m_0$, and $B = 0.5v_F^2/m_0$.

across the two critical energies. The critical energies correspond to QH plateau transitions [96]. The statistical properties of the wavefunctions near these transitions are universal [61, 97], and can be captured within a Chalker-Coddington type network model [72]. In the limit of a weak driving that couples the delocalized states, we expect the phase diagram not to depend on microscopic details. We therefore use an effective model consisting of two coupled Chalker-Coddington networks to extract the universal features of the phase diagram.

Before discussing the details of our model, it is instructive to consider the system of two coupled Chalker-Coddington networks from an RG perspective. The crossing point of the critical energies in Fig. 5 is a multicritical point that contains two kinds of relevant operators: (1) the operators that correspond to moving in energy away from criticality, related to the QH localization/delocalization transition within each individual network, and (2) the inter-layer coupling operator arising from the driving [the off-diagonal coupling in Eq. (19)].

We begin by reviewing the critical behavior of a single QH critical system/Chalker Coddington model, and then discuss the possible forms of the inter-layer coupling operator and its scaling dimension. To date, the theory for the critical point in a single QH layer has not been solved analytically. In particular, there is no exact calculation of the scaling dimension of the operator driving the QH transition. Numerically, it has been shown that the localization length, $\xi$, scales as a function of energy as

$$\xi \sim \frac{1}{|\epsilon - \epsilon_C|^\nu}, \tag{22}$$

where $\nu \sim 2.6$ is the localization length critical exponent obtained from previous studies [72,

79, 87–89, 97], and $\epsilon_C$ is the critical energy ($\epsilon_{\pm 1}$ in the previous discussion). Assuming that the QH transition is described by a scale-invariant critical theory, this critical exponent is expected to correspond to a relevant operator with scaling eigenvalue $y_\epsilon = \frac{1}{\nu}$ and scaling dimension $x_\epsilon = 2 - y_\epsilon$ [98].

Now consider two QH systems with Gaussian random potential disorder that is uncorrelated between the two systems. The simplest coupling term between the corresponding fields $f_1$ and $f_2$ is of the form

$$S_p = \int d^2 r \, \tilde{t}_p \left[ \overline{f}_1(r) f_2(r) + \overline{f}_2(r) f_1(r) \right].$$ (23)

To get the scaling dimension of this operator, one can compute the four-point correlation function

$$\overline{\langle O_t(r) O_t^\dagger(r') \rangle_\epsilon} = \overline{\langle \overline{f}_1(r) f_2(r) \overline{f}_2(r') f_1(r') \rangle_\epsilon}$$
$$= \overline{\langle \overline{f}_1(r) f_1(r') \rangle_\epsilon} \; \overline{\langle f_2(r) \overline{f}_2(r') \rangle_\epsilon}.$$ (24)

Here we have defined the "tunneling operator" $O_t(r) = \overline{f}_1(r) f_2(r)$, while $\langle \cdot \rangle_\epsilon$ denotes Grassmann integration ($\epsilon$ is the energy), and the overline denotes disorder averaging. The expectation values are evaluated with respect to the unperturbed action with $\tilde{t}_p = 0$. The second line follows from the fact that 1) $f_1$ and $f_2$ are decoupled, and therefore the Grassman integrations over $f_{1,2}$ are independent, and 2) the disorder potentials are uncorrelated. The individual correlation functions $\overline{\langle \overline{f}_{1,2}(r) f_{1,2}(r') \rangle_\epsilon}$ decay exponentially with distance [97], making the operator $O_t(r)$ irrelevant. This is because the phases of $\langle \overline{f}_{1,2}(r) f_{1,2}(r') \rangle_\epsilon$ are different for each disorder realization (and independent for the two systems 1 and 2).

Importantly, under RG, disorder can generate additional, more relevant terms. For example, to second order in $\tilde{t}_p$, a density-density coupling term of the form $O_{dd}(r) = \rho_1(r) \rho_2(r)$ (where $\rho_{i=1,2} = \overline{f}_i f_i$) can be generated. Its correlation function in the decoupled action is

$$\overline{\langle O_{dd}(r) O_{dd}(r') \rangle_\epsilon} = \overline{\langle \rho_1(r) \rho_2(r) \rho_2(r') \rho_1(r') \rangle_\epsilon}$$
$$= \overline{\langle \rho_1(r) \rho_1(r') \rangle_\epsilon} \; \overline{\langle \rho_2(r) \rho_2(r') \rangle_\epsilon}.$$ (25)

In this case, the correlation functions of the individual systems do not decay exponentially, since they do not have random phases. The correlation function (25) is expected to decay as a power law when the two QH systems are at their plateau transitions [97].

Although as of yet it is not possible to analytically compute scaling dimensions of operators in the QH plateau transition, we can determine them numerically. To this end, we now employ a network model to determine the phase diagram of the two coupled QH systems that we use to describe the emergence of the AFAI phase. We find evidence that the operator $O_{dd}$ is relevant at the multicritical point, and drives the combined system to a localized phase even if its initial amplitude (proportional to $|\tilde{t}_p|^2$) is small.

## 4.1 Network Model Description of the QH Transition

We briefly review the Chalker-Coddington network model for the QH plateau transition. The model consists of two kinds of building blocks: (1) fixed (i.e., non-random) scattering matrices parametrized by a transmission amplitude $t \in [0, 1]$ at network nodes and (2) random phase matrices with a uniform distribution over $U(1)$ along network links.

To be precise, the network model has two types of nodes, which are related by a $\frac{\pi}{2}$ rotation (see Fig. 6a). The explicit forms of the scattering matrices are:

$$S_A = \begin{pmatrix} -r & t \\ t & r \end{pmatrix}, \qquad S_B = \begin{pmatrix} -t & r \\ r & t \end{pmatrix}.$$ (26)

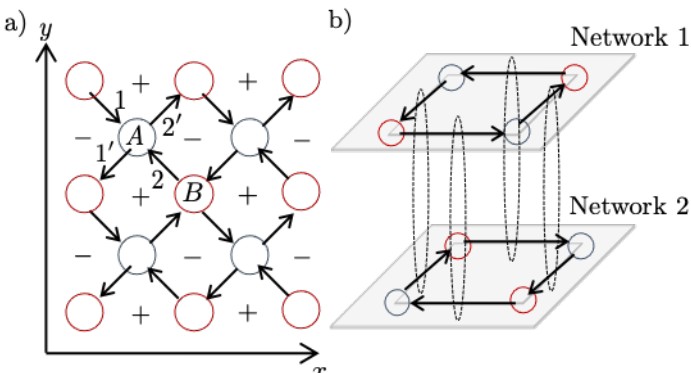

Figure 6: a) A schematic of the network model. Blue nodes host scattering matrices $S_A$ and red nodes host scattering matrices $S_B$ that are rotated by $\frac{\pi}{2}$. The links carry random $U(1)$ phases. States with $\epsilon \gtrless 0$ tend to circle around the $+/-$ plaquettes. The state with $\epsilon = 0$ is delocalized. b). In the coupled network model, we add a second layer with switched colors and opposite arrows. We also add inter-layer nodes that scatter between the two networks, indicated by the dotted loops.

These matrices relate the incoming amplitudes (1 and 2 in Fig. 6) to the outgoing amplitudes (1' and 2'), and $r = \sqrt{1-t^2}$ is the reflection amplitude. To calculate the localization length, it is convenient to use the geometry of a long cylinder. We denote the number of nodes around the circumference by $L_y$ and the number of nodes along the length by $L_x$, and we take $L_x \gg L_y$. The transport from one end of the cylinder to the other end can then be computed using transfer matrices. These relate the amplitudes to the left of a node (1 and 1') to the amplitudes to the right (2' and 2). The transfer matrices for the two types of nodes $A$ and $B$ are given by

$$T_A = \frac{1}{t}\begin{pmatrix} 1 & r \\ r & 1 \end{pmatrix}, \qquad T_B = \frac{1}{r}\begin{pmatrix} 1 & t \\ t & 1 \end{pmatrix}. \tag{27}$$

Denoting the transfer matrices for a column in the $y$ direction by $T^c_{A/B}$ (of size $2L_y \times 2L_y$) and the diagonal random phase matrices describing disorder on the links by $\Phi_{A/B,x}$, the total transfer matrix describing transport from one end of the cylinder to the other is

$$T_{QH} = \prod_{x=0}^{L_x} \Phi_{B,x} T^c_B \Phi_{A,x} T^c_A. \tag{28}$$

To see the physical meaning of the transmission amplitude $t$ in the context of the QH system, one can relate it to $\epsilon$, a quantity proportional to the energy deviation from the delocalization energy [79]:

$$t = \frac{1}{\sqrt{1 + e^{-\pi\epsilon/2}}}, \qquad r = \frac{e^{-\pi\epsilon/4}}{\sqrt{1 + e^{-\pi\epsilon}}}. \tag{29}$$

Notice that as $\epsilon \to \infty$, $t \to 1$ and as $\epsilon \to -\infty$, $t \to 0$. At $\epsilon = 0$, $t$ and $r$ are equal: $t = r = \frac{1}{\sqrt{2}}$. The network model is illustrated in Fig. 6a, with the scattering matrices $S_A$ describing the blue nodes and $S_B$ describing the red nodes. The network can be thought of as a grid of valleys ($-$ plaquettes) and summits ($+$ plaquettes). States with $\epsilon < 0$ tend to encircle the valleys, while states with $\epsilon > 0$ tend to encircle the summits. All these states are localized. At $\epsilon = 0$, the states are delocalized, and correspond to the QH critical states.

## 4.2 System of Two Coupled Network Models

We now consider two QH systems, labeled 1 and 2. The intra-layer transmission amplitudes are parametrized by $\epsilon$ and $\delta$ as follows:

$$t_1 = \frac{1}{\sqrt{1 + e^{-\pi(\epsilon + \delta/2)}}}, \qquad t_2 = \frac{1}{\sqrt{1 + e^{-\pi(\epsilon - \delta/2)}}}. \tag{30}$$

In the driven system, the $\epsilon$ axis corresponds to the quasienergy (see Fig. 2), while $\delta$ corresponds to the detuning of the driving frequency from resonance. As we explain below, to capture the opposite chirality of the critical states of the two systems, we reverse the direction of propagation on the links of system 2 relative to system 1, which also switches the $A$ and $B$ transfer matrices in system 2 (Fig. 6b).

On each link of the doubled system, we replace the random diagonal matrix $\Phi_{A/B,x}$ by

$$\Phi_{A/B,x} \rightarrow \Phi_{A/B,x,2} P^c \Phi_{A/B,x,1}, \tag{31}$$

where $P^c$ is a transfer matrix describing scattering between the two systems (originating from the driving in the original problem), as illustrated in Fig. 6b. The transfer matrix $P^c$ acts on an entire column, and is constructed from $2 \times 2$ blocks that act on the amplitudes of the two systems in a pair of links connected by the dotted ellipses in Fig. 6b. Each block is occupied by a matrix $P$ parametrized by $t_p \in [0, 1]$:

$$P = \frac{1}{\sqrt{1 - t_p^2}} \begin{pmatrix} 1 & t_p \\ t_p & 1 \end{pmatrix}. \tag{32}$$

The total transfer matrix for the system of two coupled network models of dimension $L_x \times L_y$ is given by

$$T_{L_x, L_y} = \prod_{x=0}^{L_x} \Phi_{B,x,2} P^c \Phi_{B,x,1} T_B^c \Phi_{A,x,2} P^c \Phi_{A,x,1} T_A^c. \tag{33}$$

At $t_p = 0$, we have $P = \mathbb{1}$ and the model describes two uncoupled QH systems. At the special point $t_p = \frac{1}{\sqrt{2}}$, we have $r_p = \sqrt{1 - t_p^2} = t_p$. For $t_p > \frac{1}{\sqrt{2}}$, we expect that all states become localized because electrons tend to scatter back and forth between the two networks in closed loops.

We rewrite the parameters $\epsilon, \delta$, and $t_p$ in terms of $J_\epsilon, J_\delta$, and $J_p$ [99], defined as:

$$J_\epsilon = \frac{2}{1 + e^{-\pi\epsilon}} - 1, \tag{34}$$

$$J_\delta = \frac{2}{1 + e^{-\pi\delta/2}} - 1, \tag{35}$$

$$J_p = 4t_p^2. \tag{36}$$

These new scaling parameters obey $J_\epsilon \in [-1, 1], J_\delta \in [-1, 1], J_p \in [0, 4]$. Because $J_\epsilon \propto \epsilon$ near criticality, the scaling dimension of $J_\epsilon$ should be the same as the scaling dimension of $\epsilon$. $J_\delta$ and $\delta$ are similarly related. On the other hand, $J_p$ is always positive and $J_p \propto t_p^2$, consistent with the discussion in the beginning of Sec. 4.1 where we argued that the relevant inter-layer term should be a density-density term, generated at order $\tilde{t}_p^2$.

## 4.3 Qualitative Features of the Phase Diagram

We now consider the phase diagram of the coupled two-network system as a function of $J_\epsilon, J_\delta$, and $J_p$. To begin, in order to interpret the different phases, we first comment on the meaning

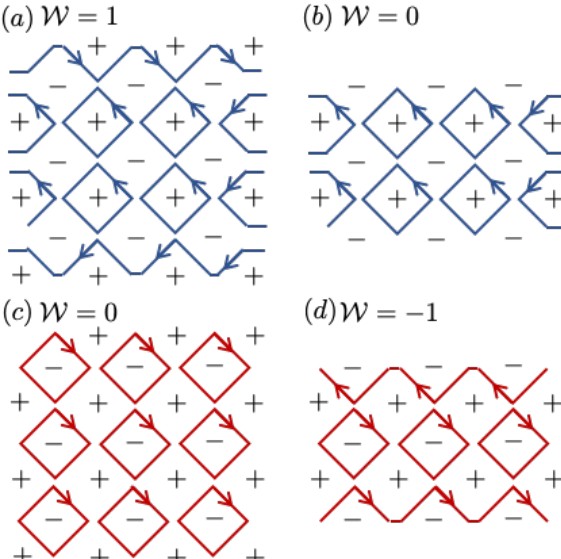

Figure 7: (a,b): Two different edge terminations for a Chalker-Coddington network model at a given energy, $\epsilon > 0$. In the termination in panel (a), the horizontal strip has a chiral edge state, whereas the termination in panel (b) does not. (c,d): The same network model with two edge terminations, at an energy $\epsilon < 0$ (across the transition). In the edge termination of (a,c), the winding number goes from 0 to 1 as $\epsilon$ increases through zero, whereas in the termination of (b,d), the winding number changes from $-1$ to 0.

of the winding number $\mathcal{W}$ in the context of a single Chalker-Coddington network model. Previous work relating the Chalker-Coddington network model to Floquet sytems defines $\mathcal{W}$ as the number of edge states when the network has open boundary conditions [83]. However, this depends crucially on the termination of the system (see Fig. 7). Depending on the edge termination, tuning the energy $\epsilon$ from $-\infty$ to $\infty$ might result in a change in the winding number from 0 to 1 or from $-1$ to 0.

We strategically choose different terminations for the two networks so that tuning $J_\delta$ through zero for $J_p = 0, J_\epsilon = 0$ results in a change in the number of edge states from 0 to 2, and thus $\mathcal{W}$ changes from 0 to 2, as in Fig. 5 (red solid lines). To see why this is the case, recall that states with energy $\epsilon \pm \frac{\delta}{2} \to \infty$ tend to encircle the $+$ plaquettes and states with $\epsilon \pm \frac{\delta}{2} \to -\infty$ tend to encircle the $-$ plaquettes. In addition, the $A$ and $B$ nodes are switched in network 2, so that the direction along each link is reversed. For $J_\delta = -1$, network 1 states, with energy $\epsilon + \frac{\delta}{2} \to -\infty$, encircle the $-$ plaquettes. On the other hand, network 2 states, with energy $\epsilon - \frac{\delta}{2} \to \infty$, encircle the $+$ plaquettes (see Fig. 8a). In the limit $J_\delta = 1$, network 1 states encircle the $-$ plaquettes and network 2 states encircle the $+$ plaquettes (see Fig. 8b). It is clear from Fig. 8 that in going from $J_\delta < 0$ to $J_\delta > 0$, the number of edge states increases by two. Next, in the limit $J_p = 4$, electrons scatter between the two networks with probability 1. The scenarios with $J_\delta = -1$ and $+1$ are illustrated in Fig. 8c and 8d, respectively, and both have a single edge state so $\mathcal{W} = 1$.

We now turn our attention to the transitions between these different phases. First consider the $J_p = 0$ plane, which describes two decoupled networks. There are critical lines at $\epsilon = \pm\frac{\delta}{2}$ or equivalently $J_\epsilon = \pm J_\delta$. On these lines, one of the two transmission amplitudes $t_1$ or $t_2$ from Eq. (30) is equal to $\frac{1}{\sqrt{2}}$, and hence the corresponding network is critical.

We expect that, going out of the $J_p = 0$ plane, there would be critical surfaces that extend from the $J_\epsilon = \pm J_\delta$ critical lines, separating the regions containing the $|J_\delta| > |J_\epsilon|, J_p = 0$ points

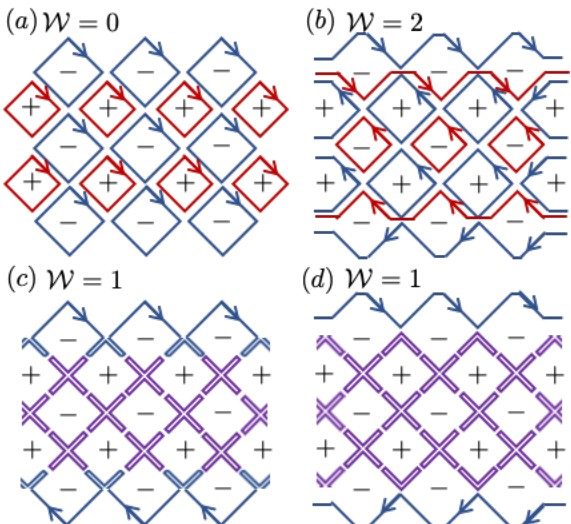

Figure 8: Phases of the coupled network model. States in network 1 are illustrated in blue and states in network 2 are illustrated in red. The edge states are easy to determine in the limit $J_\epsilon = 0$, $J_\delta = \pm 1$ and $J_p = 0$ (no inter-layer coupling, see (a) and (b)), as well as $J_\epsilon = 0$, $J_\delta = \pm 1$ and $J_p = 4$ (maximum inter-layer coupling, (c) and (d)). Going from (a) to (c) or (b) to (d) corresponds to turning on the inter-layer coupling $J_p$ at the midpoints of the links, which connects the red and blue trajectories. The purple lines indicate trajectories where states hop between the two networks in tight loops. The winding numbers corresponding to the different phases are indicated in the figure.

from the regions containing the $|J_\delta| < |J_\epsilon|$, $J_p = 0$ points (Fig. 9). In addition, as mentioned in Sec. 4.2, in the region of $t_p > \frac{1}{\sqrt{2}}$, which corresponds to $J_p > 2$, all bulk states should be localized. So all critical surfaces must lie below the plane $J_p = 2$. The numerically computed phase diagram matches well with these qualitative arguments, and is shown in Fig. 9.

## 4.4 Finite Size Scaling

In this section, we briefly review the methods used for the calculation of the localization length through finite size analysis. Identifying where the localization length diverges allows us to locate the phase transitions and thus to map out the phase diagram.

The localization length can be obtained from the product of transfer matrices. Consider $T_{L_x, L_y}$, the transfer matrix describing a system of two coupled network models (33). We define the matrix $\Gamma$ by

$$\Gamma = \lim_{L_x \to \infty} \left( T_{L_x, L_y} T_{L_x, L_y}^\dagger \right)^{\frac{1}{2L_x}}.$$  (37)

From Oseledet's theorem [86], one can show that $\Gamma$ always has positive eigenvalues, of the form $\exp(\pm \gamma_i)$, where the physical meaning of $\gamma_i$ is the exponential change in the wavefunction over a single lengthwise slice. The smallest $\gamma_i$ corresponds to the inverse of the localization length in this quasi-1D setup:

$$\xi = \frac{1}{\gamma_{\min}}.$$  (38)

The localization length is a self-averaging quantity, and is independent of the disorder realization.

We denote the reduced localization length by $\Lambda = \frac{\xi(L_y, J)}{L_y}$ with $J = \{J_\delta, J_\epsilon, J_p\}$. Then, according to one-parameter scaling, near the critical value $J = J_c$, $\Lambda$ should not separately

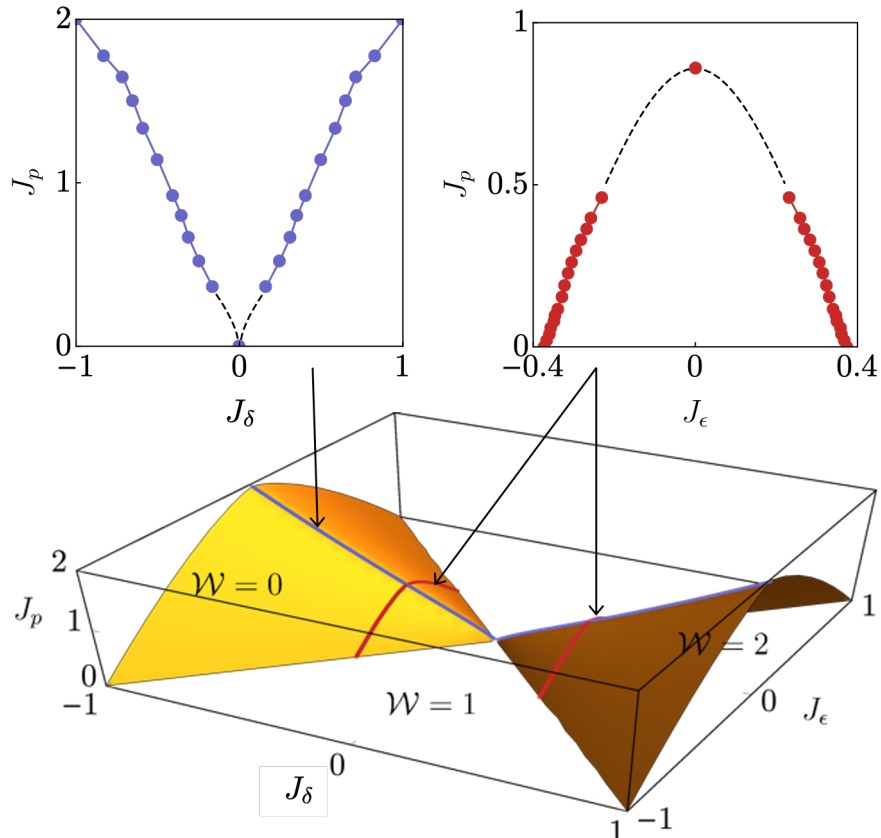

Figure 9: The phase diagram of the two coupled network system as a function of $J_\delta, J_\epsilon$, and $J_p$. The regions under the critical surfaces have $\mathcal{W} = 0, 2$ and the region above the critical surfaces has $\mathcal{W} = 1$. The insets show cross sections indicated by the blue and red lines in the 3D plot. The black dashed lines are interpolations; in particular, in the left inset, we interpolate according to $J_p \propto |J_\delta|^{2.6/4.2}$ (see the discussion in Sec. 4.5), whereas in the right inset the interpolation is parabolic.

depend on $L_y$ and $J$ but vary as $\Lambda = F\left(\frac{L_y}{\xi_\infty}\right) = \tilde{F}\left(|J - J_c| L_y^{1/\nu_J}\right)$ where we used $\xi_\infty \sim \frac{1}{|J-J_c|^{\nu_J}}$, and $F, \tilde{F}$ are universal scaling functions. Therefore, near the critical point, $\Lambda(r)$ plotted against $|J - J_c| L_y^{1/\nu_J}$ should coincide for all values of $L_y$. We therefore obtain $\nu_J$ by choosing its values such that the data for $\Lambda$ vs. $J$ collapse for different values of $L_y$. An exemplary plot showing the finite size scaling analysis and the data collapse is given in Fig. 10 for the tuning parameter $J = J_p$.

We point out that $\Lambda$ is a good choice for a scaling variable because it has a singularity in the limit of infinite system size in a localization-delocalization transition: $\Lambda \to 0$ as $L_y \to \infty$ in an insulator because $\xi_\infty$ is finite, and $\Lambda \to \infty$ as $L_y \to \infty$ in a metal. At the critical point, $\Lambda \to \Lambda_c$ for all $L_y$.

## 4.5 Results

We use the above methods to make several cross sections through the phase diagram in the $(J_\epsilon, J_\delta, J_p)$ space. The first cut is along the line $J_\delta = J_\epsilon = 0$. The resulting reduced localization length $\Lambda$ as a function of $J_p$ is shown in Fig. 10. The data are consistent with a critical point at $J_p = 0$. For an optimal value of the localization critical exponent $\nu_p \sim 4.2$, $\Lambda$ collapses with high precision onto a universal function. When $J_p = 0$, the two networks are decoupled, and deviating from lines $J_\delta = \pm J_\epsilon$ corresponds to a usual quantum Hall transition, with critical

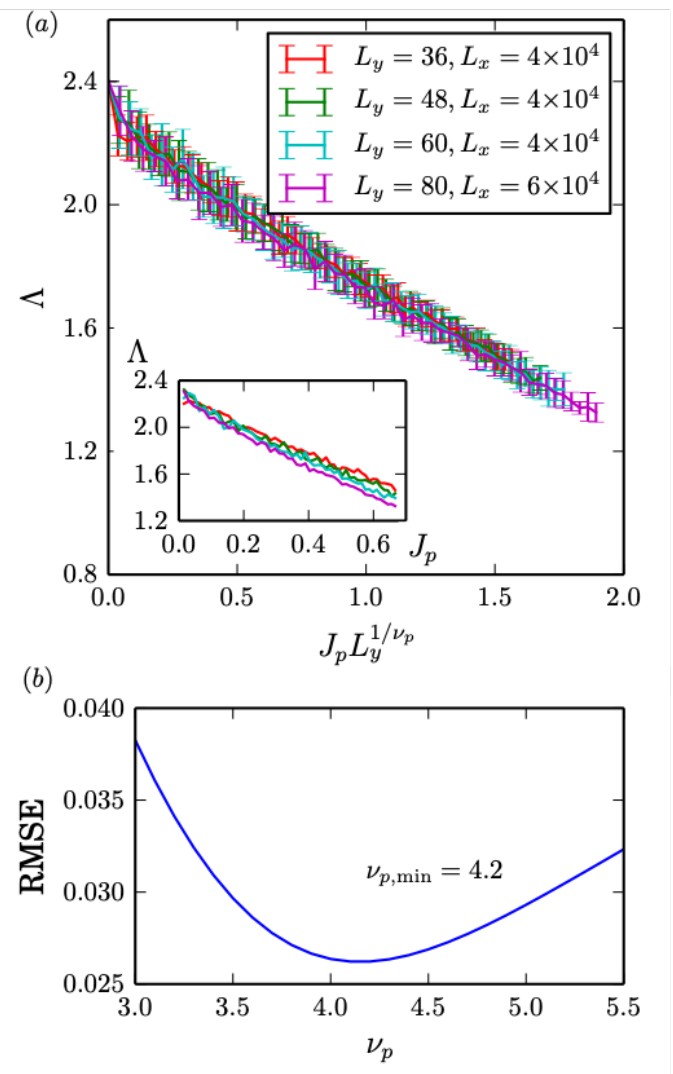

Figure 10: (a) $\Lambda$ plotted against $J_p L_y^{1/\nu_p}$ with optimal $\nu_p$ from (a). Inset: the unscaled data with the same coloring as in the main figure. (b) As discussed in the text, near the critical point, $\Lambda$ plotted against $J_p L_y^{1/\nu_p}$ should be the same for all $L_y$. Here we measure how well data from different $L_y$ match by performing a linear fit on $\Lambda(J_p L_y^{1/\nu_p})$ for $L_y = 80$, then calculating the average root mean square error (RMSE) of plots from different $L_y$ to the values given by the linear fit. The minimum RMSE is given by $\nu_p = 4.2$.

exponent $\nu \approx 2.6$ [72,87–89]. This implies that in the $J_\epsilon = 0$ plane, the critical lines should then follow the curve $J_p \propto J_\delta^{y_\delta/y_p} = J_\delta^{\nu_p/\nu_\delta} = J_\delta^{2.6/4.2}$, where $y_\delta$ and $y_p$ are the scaling eigenvalues of the corresponding operators.

The three-dimensional phase diagram (Fig. 9), is constructed using cross sections in the $J_p = 0$ plane, $J_\epsilon = 0$ plane, and $J_\delta = \pm 0.37$ planes. We point out that the lines near $J_\delta = J_p = 0$ and close to $J_\epsilon = 0$ are interpolated because the localization length around these points is very large. The points in the plane $J_\epsilon = 0$ are obtained as follows: for each value of $J_p$, a sweep through $J_\delta$ determines where $\Lambda$ peaks and collapses for all $L_y$. Because the phase diagram is symmetric with respect to $\delta \to -\delta$ and $\epsilon \to -\epsilon$, the data points only need to be obtained for one side of each cross section.

Fig. 9 implies that if the two opposite chirality states were put exactly on resonance so that $J_\delta = 0$, then any amount of coupling $J_p$ would lead to localization for all energies $J_\epsilon$. In other words, if the Floquet driving were exactly resonant with the energy $\epsilon_{-1} - \epsilon_{+1}$, then any finite coupling amplitude would cause all bulk states at all energies to localize. If the two opposite chirality states were slightly off-resonant, so that $0 < |J_\delta| \ll 1$, there needs to be sufficient coupling $J_p$ to localize the bulk states at all energies. Specifically, $J_p$ must satisfy $J_p \geq a J_\delta^{2.6/4.2}$ with the constant $a \approx 2.4$ from the numerical fit in Fig. 9a. In conclusion, we find that for any value of $J_\delta$, one can choose $J_p$ such that the line $(\mathbb{R}, J_\delta, J_p)$ does not have any critical points. Physically, this means that as long as the rotating wave approximation holds, for any amount of detuning, a sufficiently strong driving amplitude localizes all the bulk states and the system realizes an AFAI phase. Note that the network model describes the physical setup of the driven QAH system for $J_p \ll 1$. This is the weak driving limit, where the rotating wave approximation holds.

## 5   Discussion

We have shown that it is possible to obtain an AFAI by driving a QAH system at a frequency equal to the energy difference between its two delocalized states. To this end, we investigated an effective Hamiltonian for a driven QAH system, treating the effects of the Floquet drive within the rotating wave approximation. We showed that for the QAH state that is typically found in magnetically doped TIs, we can compute the resonant drive frequency using the SCBA. We argued that there should be a finite frequency window around this resonant frequency where the AFAI is stable (Fig. 5).

We then backed these arguments using a numerical study of a system of two coupled and disordered network models, which captures the universal properties of the transition. We calculated the localization length in this system and showed that the coupling operator between two delocalized states of opposite chirality is relevant, with a scaling dimension $\sim 1.8$.

Our findings demonstrate that an AFAI can be obtained out of a realistic low-energy model. Importantly, the proposed mechanism of Chern band annihilation to generate an AFAI does not depend in an essential way on the strength of disorder or on the type of driving.

We end by outlining open questions and directions for further work. As pointed out in Sec. 3, when calculating the Hall conductivity for very weak disorder, it is necessary to include the skew scattering contribution. While this has been done for a constant mass [69,70], the case of a momentum dependent mass has not been considered so far. Generically, we expect there to be some contribution because the magnetic dopants, combined with spin-orbit coupling, would be conducive to skew scattering. In the calculation, this can be accounted for by using non-Gaussian disorder. Even though this contribution may shift the critical lines slightly, we do not expect it to change the qualitative features of the phase diagram.

A natural direction for future work is to investigate how the robustness of the AFAI in the single-particle picture carries over to the interacting system. Isolated driven many-body systems are generically expected to heat up to infinite temperature. However, there is evidence that sufficient disorder can allow distinct phases to persist for long times due to many-body localization [46,100–103]. It has recently been argued that the AFAI is stable to interactions in many-body localized systems [56]. It would therefore be interesting to study the stability to interactions of the particular protocol we presented in Section 3 for realizing the AFAI.

In a solid state setup, coupling to phonons will inevitably destroy the localization at sufficiently long times [104]. However, the presence of an AFAI state may still manifest itself in interesting transient phenomena, which we leave for future study. In particular, the effect of coupling to phonons and leads on the transport signatures of a solid state AFAI (see Fig. 1) remains an important open question.

# Acknowledgements

We are grateful to M. Levin, A. Altland and S. Sondhi for helpful discussions. CZ is supported by the Kadanoff Center for Theoretical Physics at the University of Chicago and the National Science Foundation Graduate Research Fellowship under Grant No. 1746045. NL acknowledges support from the European Research Council (ERC) under the European Union Horizon 2020 Research and Innovation Programme (Grant Agreement No. 639172), and from the Israeli Center of Research Excellence (I-CORE) "Circle of Light". EB and MR acknowledge support from CRC 183 of the Deutsche Forschungsgemeinschaft (Project A01). This work was supported by a research grant from Irving and Cherna Moskowitz. MR gratefully acknowledges the support of the European Research Council (ERC) under the European Union Horizon 2020 Research and Innovation Programme (Grant Agreement No. 678862), and the Villum Foundation.

**Competing Interests:**    The authors declare no competing interests.

**Author Contributions:**    CZ performed the numerical calculations and CZ and TH performed the analytical calculations under guidance of EB, MR and NL. All authors contributed to the writing of the manuscript.

**Data Availability:**    The data of the numerical calculations are available from the authors upon reasonable request.

# A    Calculation of Hall conductivity

In this Appendix we derive Eq. (9) for the critical line in the $\Delta, \epsilon_F$ plane. At this line, the semiclassical Hall conductivity $\sigma_{xy}^{(1)}$ of species 1 (assuming that $m_1 \ll m_2$) vanishes. To simplify the notation, the superscript (1) in $\sigma_{xy}^{(1)}$ is left implicit in the remainder of this section.

We begin by discussing the different disorder contributions to the Hall conductivity. The total Hall conductivity is given by [71]

$$\sigma_{xy} = \sigma_{xy,0} + \sigma_{xy,\text{sj}} + \sigma_{xy,\text{skew}}, \tag{39}$$

where $\sigma_{xy,0}$ is the intrinsic contribution from the Berry curvature, $\sigma_{xy,\text{sj}}$ is the side-jump contribution, and $\sigma_{xy,\text{skew}}$ is the skew-scattering contribution. Both $\sigma_{xy,0}$ and $\sigma_{xy,\text{sj}}$ give contributions that are independent of the transport scattering lifetime, $\tau$, while $\sigma_{xy,\text{skew}}$ gives a contribution proportional to $\tau$. Depending on the disorder type, $\sigma_{xy,\text{skew}}$ may be zero, but if it is finite, $\sigma_{xy,\text{skew}}$ dominates the Hall conductivity in the clean limit where $\tau \to \infty$. In the following we consider systems in what was termed the intrinsic metallic regime, where $\sigma_{xy,0}$ and $\sigma_{xy,\text{sj}}$ are dominant [71]. In this regime, the Hall conductivity $\sigma_{xy} = \sigma_{xy,0} + \sigma_{xy,\text{sj}}$ is independent of the disorder strength for a fixed form of the impurity potential [105]. Specifically, in our model, the skew scattering contribution vanishes because of the symmetric (Gaussian) distribution of the disorder potential [105].

## A.1    Intrinsic Contribution

Consider the Hamiltonian describing the mode $\psi_1$, of the form

$$H(\boldsymbol{k}) = v_F(k_y \sigma_x + k_x \sigma_y) + M(k)\sigma_z, \tag{40}$$

where $M(k) = \left(m_0 - \Delta + \frac{2A^2}{m_2}\right) + Bk^2$ (where we set the off diagonal coupling terms in Eq. (19) to zero). Below we use $m \equiv m_0 - \Delta + \frac{2A^2}{m_2}$ for convenience, so $M(k) = m + Bk^2$. To evaluate $\sigma_{xy,0}$, we integrate over the Berry curvature

$$\sigma_{xy,0} = \frac{1}{\Omega} \sum_k \frac{f_k^+ - f_k^-}{(\epsilon_k^+ - \epsilon_k^-)^2} 2\mathrm{Im}[\langle u_k^- | v_y | u_k^+ \rangle \langle u_k^+ | v_x | u_k^- \rangle], \tag{41}$$

where $f_k^\pm \equiv \Theta(\epsilon_F - \epsilon_k^\pm)$ are the occupation numbers in the conduction and valence bands for the Fermi energy $\epsilon_F$, $\Omega$ is the area of the system, and $\epsilon_k^\pm = \pm\sqrt{(v_F k)^2 + (m + Bk^2)^2}$. The periodic Bloch states $|u_k^\pm\rangle$ are the $k$-dependent eigenstates of the Hamiltonian Eq. (40) defined as

$$|u_k^+\rangle = \begin{pmatrix} \cos(\theta_k/2) \\ \sin(\theta_k/2)e^{i\phi_k} \end{pmatrix}, \qquad |u_k^-\rangle = \begin{pmatrix} \sin(\theta_k/2) \\ -\cos(\theta_k/2)e^{i\phi_k} \end{pmatrix}, \tag{42}$$

where $\cos\theta_k = \frac{m+Bk^2}{|\epsilon_k|}$ and $\tan\phi_k = \frac{k_x}{k_y}$. The velocity operators $v_y$ and $v_x$ are given by

$$v_x = \frac{dH}{dk_x} = v_F \sigma_y + 2Bk_x \sigma_z,$$
$$v_y = \frac{dH}{dk_y} = v_F \sigma_x + 2Bk_y \sigma_z. \tag{43}$$

Evaluating $\mathrm{Im}[\langle u_k^- | v_y | u_k^+ \rangle \langle u_k^+ | v_x | u_k^- \rangle]$ yields

$$\mathrm{Im}[\langle u_k^- | v_y | u_k^+ \rangle \langle u_k^+ | v_x | u_k^- \rangle] = v_F^2 \cos\theta_k - 2Bv_F k \sin\theta_k. \tag{44}$$

Then, after performing the $\phi_k$ integral in (41), the Hall conductivity for $\epsilon_F$ in the conduction band becomes

$$\sigma_{xy,0} = \int_{k_F}^K dk \frac{1}{2} \frac{v_F^2 |k|(m - Bk^2)}{[(v_F k)^2 + (m + Bk^2)^2]^{3/2}}. \tag{45}$$

Due to the presence of the second Dirac field, the integration is only defined up to the second mass $m_2$, which sets the upper energy cutoff $\epsilon = m_2$ and the momentum cutoff $K \sim m_2/v_F$. Assuming that $v_F K \gg |B|K^2$, the result of the integration is

$$\sigma_{xy,0} \approx -\frac{(m + Bk_F^2)}{2\sqrt{(m + Bk_F^2)^2 + (v_F k_F)^2}}$$
$$= -\frac{1}{2}\cos\theta_{k_F}. \tag{46}$$

Importantly, this quantity is zero when $\cos\theta_{k_F} = 0$, which is true for $M(k_F) = 0$. For a constant ($k$-independent) mass term, $\sigma_{xy,0}$ is identically zero for $m = 0$ for all energies. If $m$ is replaced by $m + Bk^2$, the energy for which $\sigma_{xy,0} = 0$ has a non-trivial dependence on $m$. Notice that in order for such an energy to exist, the product $mB$ must be negative.

## A.2 Side-Jump Contribution

To account for the effects of disorder on the Hall conductivity, we must add the contributions of the diagrams in Fig. 11. We consider $\delta$-correlated random potential disorder characterized by $\langle V(r)V(r')\rangle = nV_0^2 \delta(r - r')$, with no higher moments, where $n$ is the impurity concentration. The first step is to solve for the on-shell self-energy $\Sigma(\epsilon = \epsilon_F) = \Sigma_0^R$, shown in Fig. 12. For

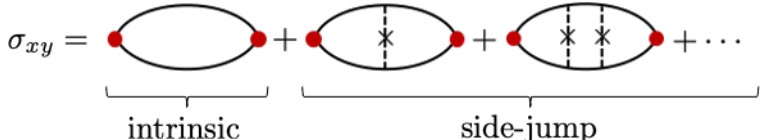

$$\sigma_{xy} = \underbrace{\phantom{XXXXX}}_{\text{intrinsic}} + \underbrace{\phantom{XXXXXXXXXXXXXX}}_{\text{side-jump}} + \cdots$$

Figure 11: Diagrams contributing to the Hall conductivity in the intrinsic metallic region. The grouping into the intrinsic and side-jump parts corresponds to the usual labels given for these processes.

the potential disorder considered here, $\Sigma_0^R$ contains two pieces, proportional to $\mathbb{1}$ and $\sigma_z$. It evaluates to

$$
\begin{aligned}
\Sigma_0^R &= -i\pi n V_0^2 \int \frac{d^2 k}{4\pi^2} G_0^R(\epsilon_F, \boldsymbol{k}) \delta(\epsilon_F - \epsilon_k^+) \\
&= -\frac{i n V_0^2}{4} \frac{\epsilon_F}{\sqrt{4B^2 \epsilon_F^2 + 4Bm v_F^2 + v_F^4}} (\mathbb{1} + \cos\theta_{k_F}\sigma_z) \\
&= -\frac{i}{4\tau_q}(\mathbb{1} + \cos\theta_{k_F}\sigma_z),
\end{aligned}
\tag{47}
$$

where we introduced the inverse quantum lifetime $(\tau_q)^{-1} = nV_0^2 \epsilon_F / \sqrt{4B^2 \epsilon_F^2 + 4Bm v_F^2 + v_F^4}$. In the limit $B \to 0$, the inverse lifetime reduces to its form in the constant mass case, i.e., $(\tau_q)^{-1} = nV_0^2 \epsilon_F / v_F^2$. Using this self energy, the SCBA Green's function becomes

$$
\begin{aligned}
G^R(\epsilon = \epsilon_F, \boldsymbol{k}) &= \frac{1}{1/G_0^R - \Sigma^R} \\
&= \frac{\epsilon_F + i\Gamma + v(k_y \sigma_x + k_x \sigma_y) + (m + Bk^2 - i\Gamma_1)\sigma_z}{(\epsilon_F - \epsilon_k^+ + i\Gamma^+)(\epsilon_F - \epsilon_k^- + i\Gamma_-)},
\end{aligned}
\tag{48}
$$

with $\Gamma = (4\tau_q)^{-1}, \Gamma_1 = \Gamma\cos\theta_{k_F}$, and $\Gamma_\pm = \Gamma(1 \pm \cos^2\theta_{k_F})$.

The recursion relation for the velocity vertex in the ladder approximation is defined by

$$
\begin{aligned}
\Upsilon_x(\epsilon = \epsilon_F, \boldsymbol{k}) &= \frac{\partial H}{\partial k_x} + nV_0^2 \int \frac{d^2 k'}{(2\pi)^2} G^R \Upsilon_x G^A \\
&= v_F \sigma_y + 2Bk_x \sigma_z + nV_0^2 \int \frac{d^2 k'}{(2\pi)^2} G^R \Upsilon_x G^A.
\end{aligned}
\tag{49}
$$

Here, the Green's functions and the vertex functions are taken at $\epsilon = \epsilon_F$. To solve this equation for $\Upsilon_x$, we decompose $\Upsilon_x = c_0\sigma_0 + c_x\sigma_x + c_y\sigma_y + c_z\sigma_z + 2Bk_x\sigma_z$, multiply by Pauli matrices

$$\Sigma^R = \qquad$$ 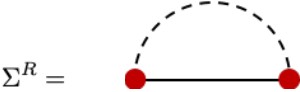

Figure 12: The diagram is constituting the lowest order contribution to the quasiparticle self-energy.

from the left, and take the trace. This gives

$$2c_i = 2v_F \delta_{i,y} + A_{ij}c_j + B_{iz},\tag{50}$$

$$A_{ij} = nV_0^2 \int \frac{d^2k'}{(2\pi)^2} \text{Tr}\left[\sigma_i G^R \sigma_j G^A\right],\tag{51}$$

$$B_{iz} = nV_0^2 \int \frac{d^2k'}{(2\pi)^2} \text{Tr}\left[\sigma_i G^R (2Bk'_x \sigma_z) G^A\right].\tag{52}$$

We express the above equation in matrix form

$$\begin{pmatrix} c_0 \\ c_x \\ c_y \\ c_z \end{pmatrix} = (2 - \mathbf{A})^{-1} \begin{pmatrix} B_{0z} \\ B_{xz} \\ 2v_F + B_{yz} \\ B_{zz} \end{pmatrix},\tag{53}$$

where the matrix $\mathbf{A}$ has the elements $A_{ij}$ defined above. Notice that $2 - \mathbf{A}$ is block diagonal, so we can diagonalize each block separately:

$$\begin{pmatrix} c_x \\ c_y \end{pmatrix} = (2 - \mathbf{A}_{xy})^{-1} \begin{pmatrix} B_{xz} \\ 2v_F + B_{yz} \end{pmatrix},\tag{54}$$

$$\begin{pmatrix} c_0 \\ c_z \end{pmatrix} = (2 - \mathbf{A}_{0z})^{-1} \begin{pmatrix} B_{0z} \\ B_{zz} \end{pmatrix}.\tag{55}$$

In order to evaluate $\mathbf{A}$, we assume that the poles of the advanced and retarded Green's functions are well separated, so that one can take the residue in either without affecting the other (see, e.g., Ref. [70]), with the result up to $\mathcal{O}(\Gamma/\epsilon_k)$ being

$$A_{x,x} = A_{y,y} = \frac{\sin^2 \theta_{k_F}}{(1 + \cos^2 \theta_{k_F})} - \frac{i\Gamma \sin^2 \theta_{k_F}}{\epsilon_F},\tag{56}$$

$$A_{x,y} = -A_{y,x} = \frac{4\Gamma \cos \theta_{k_F}}{(1 + \cos^2 \theta_{k_F})\epsilon_F}.\tag{57}$$

We point out that there is an extra term in $A_{x,x} = A_{y,y}$ of order $\frac{\Gamma}{\epsilon_F}$ which was neglected in Ref. [70]. This term does drop out in the final result for the Hall conductivity upon taking $\Gamma \to 0$. We also find

$$B_{xz} = \frac{\Gamma}{\epsilon_F \cos(\theta_{k_F})} B_{yz} = \frac{\sin^2 \theta_{k_F}}{1 + \cos^2 \theta_{k_F}} \frac{B\Gamma}{v_F},\tag{58}$$

$$B_{0z} = B_{zz} = 0.\tag{59}$$

Because $B_{0z} = B_{zz} = 0$, Eq. (55) immediately gives $c_0 = c_z = 0$. Solving Eq. (54) for $c_x$ and $c_y$ gives

$$c_x = \frac{8(1 + \cos^2 \theta_{k_F})\cos \theta_{k_F}}{(1 + 3\cos^2 \theta_{k_F})^2} \frac{v_F \Gamma}{\epsilon_F} + \frac{\sin^2 \theta_{k_F}(1 + 7\cos^2 \theta_{k_F})}{(1 + 3\cos^2 \theta_{k_F})^2} \frac{B\Gamma}{v_F},\tag{60}$$

$$c_y = \frac{2v_F(1 + \cos^2 \theta_{k_F})}{1 + 3\cos^2 \theta_{k_F}} + \frac{\sin^2 \theta_{k_F} \cos \theta_{k_F}}{1 + 3\cos^2 \theta_{k_F}} \frac{\epsilon_F B}{v_F},\tag{61}$$

where the term of order $\Gamma/\epsilon_F$ in $c_y$ was already dropped as the contraction with the Green's functions in the evaluation of the Hall conductivity means that such a term would give order

$\Gamma/\epsilon_F$ contributions to the Hall conductivity. In all other components, this step has to be deferred until the end.

We are now in the position to compute the Hall conductivity from Eq. (8) of the main text. As shown in Appendix B of Ref. [70], this breaks into an intrinsic contribution ($\sigma_{xy,0}$) from below the Fermi level and two terms $\sigma_{xy}^a$ and $\sigma_{xy}^b$ from near the Fermi level. These two terms correspond to the Fermi-surface pieces from the integrals involving $G^R G^A$ and $G^A G^A$, respectively. Since $\sigma_{xy,0}$ was already evaluated in Eq. (46), we only need to study the remaining contribution from $\sigma_{xy}^a + \sigma_{xy}^b$.

The resulting expression can be simplified slightly by observing that $\sigma_{xy}^b$ does not receive any important renormalizations from the full vertex $\Upsilon_x$, so that it can be replaced by the bare velocity operator [70]. Therefore, the equation for $\sigma_{xy}^a(\omega) + \sigma_{xy}^b(\omega)$ reads

$$
\begin{aligned}
\sigma_{xy}^b(\omega) + \sigma_{xy}^a(\omega) = &\, i\mathrm{Tr}\int \frac{d^2k}{(2\pi)^2}\left(-\frac{\partial f}{\partial \epsilon_k}\right)\times \\
&\left[-(v_F\sigma_y + 2Bk_x\sigma_z)\frac{\epsilon_k + \omega + v_F(k_x\sigma_y + k_y\sigma_x) + M(k)\sigma_z}{2\omega\epsilon_k}\right. \\
&\times (v_F\sigma_x + 2Bk_y\sigma_z)\frac{\epsilon_k + v_F(k_x\sigma_y + k_y\sigma_x) + M(k)\sigma_z}{2\epsilon_k} \\
&+ \Upsilon_x\frac{\epsilon_k + \omega + v_F(k_x\sigma_y + k_y\sigma_x) + M(k)\sigma_z + i\Gamma - i\Gamma_1\sigma_z}{2(\omega + 2i\Gamma_+)\epsilon_k} \\
&\times (v_F\sigma_x + 2Bk_y\sigma_z) \\
&\left.\times \frac{\epsilon_k + v_F(k_x\sigma_y + k_y\sigma_x) + M(k)\sigma_z - i\Gamma + i\Gamma_1\sigma_z}{2\epsilon_k}\right].
\end{aligned}
\tag{62}
$$

Substituting $\Upsilon_x = c_x\sigma_x + c_y\sigma_y + 2Bk_x\sigma_z$ and dropping terms that are either odd in momenta or traceless results in

$$
\sigma_{xy}^a + \sigma_{xy}^b = \int \frac{d^2k}{(2\pi)^2}\left(-\frac{\partial f}{\partial \epsilon_k}\right)\left[\frac{v_F^2(m - Bk^2)}{4\epsilon_k^2} - \frac{c_y v_F(2m + Bk^2)}{4\epsilon_k^2(1 + (m + Bk^2)^2/\epsilon_k^2)}\right.
$$
$$
\left. - \frac{c_x v_F(\epsilon_k^2 - m^2 + B^2k^4)}{8\epsilon_k^2\Gamma(1 + (m + Bk^2)^2/\epsilon_k^2)} - \frac{Bv_F^2 k_x^2}{\epsilon_k^2}\frac{\Gamma}{\Gamma_+}\right].
\tag{63}
$$

Plugging in $c_x$ and $c_y$ from Eq. 60, this finally yields

$$
\sigma_{xy,0} + \sigma_{xy}^a + \sigma_{xy}^b = -\frac{4\cos\theta_{k_F}(1 + \cos^2\theta_{k_F})}{(1 + 3\cos^2\theta_{k_F})^2} + \sigma_{xy}^B.
\tag{64}
$$

The first term in Eq. 64 depends only on $B$ implicitly through the definition of $\theta_{k_F}$. The second term contains $B$ explicitly and is given by

$$
\sigma_{xy}^B = -\frac{B\epsilon_F\sin^2\theta_{k_F}}{4v^2(1 + \cos^2\theta_{k_F})}\frac{8B\epsilon_F\cos^3\theta_{k_F}\sin^2\theta_{k_F} + (5 + 34\cos^2\theta_{k_F} + 41\cos^4\theta_{k_F})v^2}{(1 + 3\cos^2\theta_{k_F})^2(2B\epsilon_F\cos\theta_{k_F} + v^2)}.
\tag{65}
$$

Eq. (64) matches with the $B = 0$ result from Ref. [70]. For $M(k_F) = 0$, which is the condition for the critical line in the clean system, this simplifies to $\sigma_{xy}^B = -\frac{5B\epsilon_F}{4v_F^2}$. Physically, this means that a finite $B$ enters in Eq. (64) not only through the changes to the dispersion but also in the form of $\sigma_{xy}^B$ due to the changes of the velocity operator. This latter dependence is what renormalizes the phase transition line non-perturbatively in disorder strength.

## A.3 Disorder Effects on the Delocalization Line

We now discuss the effects of $\sigma_{xy}^B$ on the delocalization line, where $\sigma_{xy} = 0$. To do this, we expand Eq. (64) in powers of $\cos\theta_{k_F}$, yielding

$$\sigma_{xy} = -4\cos\theta_{k_F} - \frac{5B\epsilon_F}{4v_F^2} + \mathcal{O}\left(\cos^2\theta_{k_F}\right). \tag{66}$$

This expansion is justified if the contribution from $\sigma_{xy}^B$ does not shift the delocalization lines far from the $B = 0$ lines, which occur at $\cos\theta_{k_F} = 0$. This is true when $\frac{B\epsilon_F}{v_F^2} \ll 1$. Substituting the generic mass variable $m$ by the definition used in the main text, $m \to m_0 - \Delta + \frac{2A^2}{m_2}$, and dropping terms higher than first order in $\frac{B\epsilon_F}{v_F^2}$, one obtains

$$\Delta_c \approx m_0 + \frac{2A^2}{m_2} + \frac{21}{16}\frac{B\epsilon_F^2}{v_F^2}. \tag{67}$$

Interestingly, the curvature of this critical line $\Delta_c(\epsilon_F)$ is somewhat larger than the value it would take if the contribution from $\sigma_{xy}^B$ were ignored (which would be given by $\Delta_c = m_0 + \frac{2A^2}{m_2} + \frac{B\epsilon_F^2}{v_F^2}$ at small $\epsilon_F$).

In summary, we computed the Hall conductivity with intrinsic and side jump contributions for a 2D Dirac-like system with a small quadratic perturbation in $\mathbf{k}$. The condition satisfied by the critical line, where $\sigma_{xy}^{(1)} = 0$ (the Hall conductivity for the field $\psi_1$ in Eq. (3)), is given by Eq. (67).

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
