# Peer review of "Proposal for realizing anomalous Floquet insulators via Chern band annihilation"

_SciPost Physics, doi:SciPost Phys. 12, 124 (2022)_

## Round 2 · Referee Report · Anonymous · 2022-1-6

Strengths

1-Connect the interesting theoretical proposal of an anomalous Floquet Anderson insulator to possible materials-based platforms
2-Clearly written with an important conclusion that is well-substantiated by the complementary methods of Hamiltonian-based formulation and a network model
3-Well organized with the physical picture delivered up front, followed by a pedagogical elaboration on the calculation details which can serve as an excellent reference for future works performing a similar analysis

Weaknesses

1-The models are ultimately phenomenological in nature, and so are not directly applicable to any specific materials (beyond the general expectation of a disordered, magnetically doped TI). In addition, as discussed in the paper the effects of electron-electron interactions and phonons are left for future studies. In any case, addressing these are obviously beyond the scope of the present work, and does not in any way diminish the value of the manuscript.

Report

In this work, the authors demonstrate how a (nearly) resonantly driven, disorders quantum anomalous Hall insulator could realize the anomalous Floquet Anderson insulator (AFAI), in which all the bulk states are localized but the chiral Floquet edge state remains. While concrete models of such AFAI have bene proposed, they were restricted to the scope of exceptionally well-controlled (and, in the near-term, small-scale) quantum simulators. It is an interesting and important question to ask if such Floquet topological phases could be realized in a materials-based solid-state platform. The present paper attacks this problem based on the intuition that in a disordered quantum anomalous Hall system most of the bulk states are localized, except for the topologically guaranteed delocalized state carrying the nontrivial Chern number. This then suggests there could be a resonant frequency for which delocalized states with opposite Chern numbers could be coupled and mutually annihilated. The physical picture is substantiated by careful analysis based on complementary approaches, and the authors convincingly argued that there should be a window of nearly resonant frequency in which an arbitrarily weak driving could stabilize the AFAI phase. As such, I believe this work “opens a new pathway in an existing or a new research direction, with clear potential for multipronged follow-up work,” and therefore meets the acceptance criteria of SciPost Physics.

Requested changes

The paper is already well-written and no changes are requested by this referee

---

## Round 2 · Referee Report · Anonymous · 2022-1-27

Strengths

1- Well-written and pedagogically introduced main idea as well as calculations for proof-of-principle.
2-Topically addressing a highly interesting anomalous topological phase that arises only under periodic driving; would appeal to a relatively wide range of audience from theory and experiments.
3-Proposing a realistic way for the realization of such an exotic phase.

Weaknesses

1-Does not cover the full range of problems and parameter regimes a real experimental implementation would face.

Report

The Authors propose and theoretically study a scheme to realize the relatively-recently predicted anomalous Floquet Anderson insulators. This topologically nontrivial phase involves a chiral edge state present at all (quasi)energies and crossing the entire Floquet Brillouin zone while having all bulk states localized, as opposed to the equilibrium case which requires some delocalized bulk states. Current proposal is based on a simple and elegant idea of starting with a Chern insulator that supports edge states and bulk bands with opposite Chern numbers. Assuming weak disorder to facilitate AFAI, they resonantly couple the Chern bands with a periodic drive which results in opposite Chern bands annihilating each other while leaving the chiral edge state behind. Different complementary analysis and models have been employed and detailed sufficiently. The approximations they made are clearly stated. Although it relies on calculating the delocalization energies and hence considers simple models, the proposed method holds as proof-of-principle as the resonance condition is also relaxed at the end and should be able to applied to more complex and realistic settings involving phonons, interactions etc. I suggest publication in SciPost Physics as it relates to a ground-breaking theoretical discovery in sufficient detail and opens a new pathway for the observation of an exotic AFAI phase.

Requested changes

1-Although they are clearly careful in the introduction and elsewhere about the nature of their theoretical proposal, the title can still be mistaken as a true experimental realization of an anomalous Floquet Anderson phase. It would be useful to amend this.

2-It could be possible to realize an AFAI phase by using alternative schemes or systems as also mentioned in the paper. The paper would benefit from a clearer contrast regarding where the current predictions stand with respect to e.g. Ref.65 which focuses more on disorder effects.

---

## Round 3 · List of Changes

We added "Proposal for" at the beginning of the title to emphasize that this paper is a theoretical proposal, not an experimental result. We also added a paragraph at the top of pg 3 ("Note that while we study Chern band...") to clarify the relationship between our work and another proposal for realizing the AFAI (Ref. 65), based on tuning disorder.

You are currently on this page

Resubmission 2108.01708v3 on 9 February 2022

---

## Editorial Decision

published